# Reactivation of a Hospital-Based Therapy Dog Visitation Program during the COVID-19 Pandemic

**DOI:** 10.3390/ani12141842

**Published:** 2022-07-20

**Authors:** Lisa Townsend, Jennifer K. Heatwole, Nancy R. Gee

**Affiliations:** 1Department of Pediatrics, Division of Adolescent Medicine, School of Medicine, Virginia Commonwealth University, Richmond, VA 23298, USA; 2Department of Psychiatry, School of Medicine, Virginia Commonwealth University, Richmond, VA 23298, USA; nancy.gee@vcuhealth.org; 3Children’s Hospital of Richmond (CHoR), Richmond, VA 23298, USA; 4Center for Human-Animal Interaction, Virginia Commonwealth University, Richmond, VA 23298, USA; heatwolejk@vcu.edu

**Keywords:** therapy dog visitation, hospital, COVID-19, healthcare workers, safety protocols, animal welfare

## Abstract

**Simple Summary:**

Negative mental health outcomes have affected healthcare workers, patients, and community members following pandemics: most recently, the SARS-CoV-2 (COVID-19) outbreak. Although therapy dog visitation programs are associated with reduced stress, most hospital-based programs were placed on hiatus during the COVID-19 pandemic. This study examined human–animal interactions during the reactivation of a hospital-based therapy dog program during the pandemic. Characteristics of the interactions and the participants involved were recorded and analyzed. Findings indicated that most visit recipients were healthcare workers, while the longest interaction times occurred with adult and pediatric patients. High levels of adherence to human and animal safety protocols indicate that human–dog therapy teams can safely return to hospital visitation work.

**Abstract:**

This study examined human–animal interactions during the reactivation of a hospital-based therapy dog program during the COVID-19 pandemic. Data were collected from human–dog interactions at an academic medical center in Virginia. Interaction length, participant role, age group (pediatric or adult), and observed gender were recorded. Handler adherence to human and animal safety protocols (donning personal protective equipment (PPE), using hand sanitizer, and limiting visit length) was measured. Observations from 1016 interactions were collected. *t*-tests and analysis of variance were conducted. Most visit recipients were healthcare workers (71.69%). Patients received longer visits than other participants (F(4880) = 72.90, *p* = <0.001); post hoc Bonferroni analyses (*p* = 0.05/4) showed that patients, both adult (M = 2.58 min, SD = 2.24) (95% C.I = 0.35–1.68) and pediatric (M = 5.81, SD = 4.38) (95% C.I. 3.56–4.97), had longer interaction times than healthcare workers (M = 1.56, SD = 1.92) but not visitors (*p* = 1.00). Gender differences were not statistically significant (t(552) = −0.736), *p* = 0.462). Hand sanitizer protocols were followed for 80% of interactions. PPE guidelines were followed for 100% of visits. Most interactions occurred with healthcare workers, suggesting that therapy dog visits are needed for this population. High adherence to COVID-19 safety protocols supports the decision to reactivate therapy animal visitation programs in hospitals. Challenges to safety protocol adherence included ultra-brief interactions and crowds of people surrounding the dog/handler teams. Program staff developed a “buddy system” mitigation strategy to minimize departures from safety protocols and reduce canine stress.

## 1. Introduction

Hospital-based therapy dog programs provide important relief from physical and emotional discomfort for many types of patients. Hospitalization is often associated with anxiety, discomfort, loneliness [1,2,3], and unpleasant or distressing sensory experiences [4]. These experiences may increase vulnerability to anxiety, depression, and reductions in well-being [5]. Therapy dog visits have been associated with improvements in ratings of stress, anxiety, fear [5], pain, depression, well-being [6], loneliness, and boredom [4,5]. Interacting with a dog has also been associated with improvements in certain physiological parameters, such as blood pressure, heart rate, cardiovascular reactivity, exercise, and motor functioning [4,5,7,8]. Benefits have been demonstrated among adult [9] and child inpatients [10].

The hospital environment may also place strain on healthcare workers, who are vulnerable to poor mental health outcomes [11] due to job-specific stressors such as long shifts with heightened psychological demands. Compassion fatigue and burnout are common sequelae of healthcare work [12]. Physician burnout has been reported in 55–70% of emergency healthcare workers and 45–50% of non-emergency workers [13,14,15]. This is not only concerning for the well-being of healthcare workers themselves, but also has implications for patient care. Decreases in empathy and compassion for patients have been associated with healthcare worker stress; furthermore, chronically stressed nurses are more likely to make medical errors [16].

The recent COVID-19 pandemic has heightened existing psychological demands on healthcare workers. During the pandemic, healthcare workers have experienced sleep disturbances and insomnia [17,18,19,20,21], lack of personal protective equipment (PPE) [17,19,22], burnout, and mental exhaustion [17,23]. Healthcare workers may experience trauma (or vicarious traumatization) by watching patients suffer or pass away on a frequent basis [17]. The combination of these major and frequent stressors has increased the prevalence of PTSD [24] and suicidal thoughts and behaviors among healthcare workers [17,20,22].

Hospital-based therapy dog programs have been shown to alleviate stress among healthcare workers. Barker [25] found that 5 min with a therapy dog produces the same amount of stress reduction as 20 min of quiet rest. A study by Kline [26] revealed that healthcare workers rate their stress levels lower after spending 5 min with a therapy dog than following 5 min of coloring. Jensen [11] showed that healthcare workers reported less work-related burnout, less job-related depression, and less intention to leave one’s job after interacting with a therapy dog. A recent systematic review including 12 studies suggested that it is feasible to implement such programs in healthcare settings and that they may be associated with reductions in healthcare worker stress [27]. Due to therapy dogs’ unique ability to provide significant stress relief within relatively short periods of time, hospital-based therapy dog programs could provide significant amelioration of healthcare worker stress during and following the pandemic.

### 1.1. Barriers to Program Reactivation

Concerns regarding COVID-19 transmission have augmented existing concerns about infection prevention for therapy animal visits in healthcare settings. Consequently, most hospital-based therapy animal visitation programs were suspended during the height of the pandemic, a time when the benefits of such visits may have been sorely needed. Some programs, such as the PAWS Your Stress Therapy Dog Program of the University of Saskatchewan and St. John Ambulance, transitioned to an online format, where therapy dog visits were conducted virtually [28]. Although early findings indicate that such online programs are well-received and important sources of social connection [28], no data exist as yet to support the comparative effectiveness of virtual vs. in-person visits in their effects on patients and healthcare workers.

As knowledge accumulated regarding the SARS-CoV-2 virus, the Centers for Disease Control and Prevention (CDC) issued a statement that animals do not significantly contribute to the spread of COVID-19 [29], especially if standard and pandemic-specific infection prevention protocols are followed. For example, face masks and eye protection can significantly reduce the transmission of airborne infections. This finding altered the risk landscape for hospital and program administrators, paving the way for hospital-based therapy animal programs to consider reactivation. Programs have successfully managed risk for transmission of other common infections, such as methicillin-resistant staphylococcus (MRSA), by implementing appropriate infection control protocols (such as the use of hand sanitizer before and after touching a dog) [30]. However, there are no data on infection control protocol adherence among human–animal therapy dog couplets in hospitals during the COVID-19 pandemic.

### 1.2. Animal Welfare during Reactivation

Reactivation during the pandemic offered a unique opportunity to examine the impact of program reactivation on aspects of canine welfare. The program’s therapy dogs had been on hiatus from hospital visitation for a year, and significant changes were made to how humans navigated the hospital environment. For example, only four people were allowed in an elevator car at once, which caused bottlenecks and longer wait times in the main hallways. Patients, visitors, and staff wore masks and face shields or goggles that altered their appearance and non-verbal social cues. One way of reducing canine stress during reactivation was to adhere strictly to visit time limits [4,10], which can vary greatly depending upon program and setting [31]. Furthermore, individual interactions that take place during visits can vary greatly in time, activity (such as petting vs. talking to a dog), and location within the hospital [30,31]. COVID-19 and associated risk reduction strategies may change the way people interact with dogs—for example, more people interacting at one time or how they appear to the dog. There are no data that characterize the behavior of therapy dogs, handlers, and visit recipients in a hospital setting during the pandemic. Detailed exploration of therapy dog program delivery in a hospital during a pandemic may help similar institutions to make well-informed decisions about implementation that promote human and animal welfare.

### 1.3. Purpose of Current Study

The COVID-19 pandemic offered a unique opportunity to examine characteristics of human–animal interactions in a hospital setting as teams were reactivated and implementing new safety protocols. Detailed examination of human–animal interaction characteristics and the implementation of infection prevention protocols in a hospital-based therapy animal program can inform program development and improvement efforts for all hospital-based AAI programs. Furthermore, the variability in the execution of human–animal interaction programs in hospitals makes it difficult to effectively draw consistent/generalizable conclusions about canine-assisted interactions (CAIs) in a hospital setting [30]. Interactions vary in time, location, and frequency, and visit recipients differ along a wide range of characteristics. In addition, there is significant variation in infection prevention protocols and adherence to them. Although the literature contains examples of model programs and protocols [4], there are few finely grained descriptions of human–animal interactions inside a hospital, particularly during the global COVID-19 pandemic. Information regarding hospital-based therapy dog interactions and the implementation of infection prevention protocols can provide essential information regarding meeting service needs while maintaining appropriate safety precautions during a global pandemic.

This study aimed to provide a behaviorally based description of canine-assisted human–animal interactions among a large sample of visit recipients inside a hospital during a global pandemic. By doing so, the authors hope to provide data that inform decision-making about the deployment of therapy dog services, challenges to safety protocol implementation, and strategies for program improvement. This study examined characteristics of human–animal interactions and adherence to human and animal safety and welfare protocols during the reactivation of a hospital-based therapy dog visitation program. This information may also be helpful in future situations that may require hospital administrators to decide whether to put a program on hiatus or keep it active. The protocol was deemed exempt from review by the university Institutional Review Board and the Institutional Animal Care and Use Committee. Exemption criteria were met given the quality assurance purpose of the study and that no data were collected directly from visit recipients or animals.

## 2. Materials and Methods

### 2.1. Therapy Dog/Handler Teams

Dogs on Call is a therapy dog visitation program established in the Center for Human–Animal Interaction at Virginia Commonwealth University (VCU) School of Medicine in 2001. Each Dogs on Call team consists of one dog and one handler. All handlers must provide documentation of external therapy dog registration (Pet Partners or Alliance of Therapy Dogs). Handlers must also complete VCU Medical Center volunteer services training (such as a background check and HIPAA education), Dogs on Call training, and adhere to the center’s policies and procedures, including human health screenings and vaccinations. In total, 20 handlers and 20 dogs were observed during the execution of this study. Table 1 details the standard (pre-COVID-19) and enhanced (during COVID-19) health requirements and safety protocols for handlers and their dogs.

#### Handler and Therapy Dog Characteristics

Handlers are routinely asked to provide demographic information about themselves and their dog(s) for administrative purposes to the Center for Human–Animal Interaction. These data were accessed for the dog/handler teams that participated in this quality assurance investigation and are presented in the results below.

### 2.2. Measures

#### 2.2.1. Participant Role

Participants were individuals in the hospital who interacted with the dog/handler teams. Participant roles were classified as adult patient (a hospitalized person visibly over the age of 18), pediatric patient (a hospitalized person visibly under the age of 18), public adult (any person not employed by the hospital or receiving treatment who was visibly over the age of 18), public child (any person not employed by the hospital or receiving treatment who was visibly under the age of 18), or healthcare worker (HCW). Determining whether a patient was an adult versus a child was facilitated by the location of the patient in the hospital, because children are typically treated in pediatric units. Public adult or public child status was determined by the absence of an employee badge and/or uniform or the presence of either a visitor wristband or a visitor name tag. HCWs were defined as any person employed by VCU Health as indicated by a badge depicting the VCU Health logo, staff member name, and department. HCWs included nurses, doctors, social workers, administrative faculty, maintenance workers, and medical students as well as volunteer services (VS) staff. 

#### 2.2.2. Observed Gender

Observed gender data collection began on the 24th visit. The term “observed gender” is used because there was no way for the researchers to confirm individuals’ gender identity without asking them directly. Participants were defined as “male” if they displayed a traditionally masculine appearance and “female” if they presented a traditionally feminine appearance. 

#### 2.2.3. Total Visit Time

The observer started a timer at the beginning of the visit (when the Dogs on Call team opened the door to walk into the VS office). The timer continued to run as teams interacted with people in the hospital and was stopped when teams left VS (the door closed) after checking out. Total visit time was recorded for each visit. The timer was located at the top of the researcher’s clipboard so that times could be noted at a glance. 

#### 2.2.4. Time Spent in Volunteer Services (VS)

The time spent during volunteer check-in and check-out was recorded; check-in start time began when the door to VS opened and ended when the door closed and hospital visitation began. The same recording strategy was used during check-out. Total check in and check out times were added together to determine the total time spent in the VS office.

#### 2.2.5. Interaction Characteristics

Interaction characteristics were recorded on a pre-defined checklist on which the researcher made tick marks (see Figure A1). This checklist was developed by the authors based on the human–animal interaction expertise of the third author (N.R.G.) and feedback from handlers regarding their experiences in the hospital. Behavior was classified as an interaction if a person engaged with a therapy dog for four seconds or more and paused to visit the handler or dog. Four seconds was used as the cut off based on preliminary casual observations of a subset of interactions that our team labeled “drive-by” interactions in which an individual would walk by and run their hand along the dog’s body as the dog passed by but did not stop and spend time engaging in an interaction with the dog/handler. Interactions were classified in one of three categories: (1) Talk; a person talking to a dog, (2) Pet; petting/touching a dog, or (3) Talk and Pet; talking while touching and petting the dog. Totals of Talk, Pet, or Talk and Pet interactions were recorded. The observer recorded how long each individual interaction took by looking at the running timer attached to a clipboard. When the first interaction began, the observer recorded the start time. End time was recorded when the subject was no longer talking to or petting the dog. *Total interaction time* was later calculated in seconds.

It was possible for multiple people to talk and/or pet the dog at the same time, meaning that multiple people could participate in one interaction. The *total number of people* involved in each interaction was recorded. *Participant role* was recorded for each person who interacted with the dog–handler team. As described above, roles were defined as adult patient, pediatric patient, public adult, public child, HCW, or a member of the volunteer services staff. Groupings of people from multiple roles were defined as a “mixed” population. Each interaction was coded as taking place with either male (only males participated in the interaction), female (only females participated in the interaction), or mixed group (both males and females participated in the interaction). *Observed Gender Total* was used to obtain a running total of males and females who participated in the interactions. Observed gender was used as a categorical variable to examine gender differences in interaction characteristics. 

#### 2.2.6. “Love Bombing”

The term “love bombing” was developed by the authors following initial feedback from handlers as they returned to hospital visitation. It was defined as an interaction consisting of three or more people that created crowding. Crowding was coded positively when a team’s ability to move throughout the hospital was impeded by the number of people present during an interaction. Each interaction was coded as “yes” if the interaction met qualifications of a love bomb or “no” if the interaction did not meet those criteria. The number of people who participated in a love bomb was also recorded.

#### 2.2.7. Floor

Visits took place on various floors of the VCU Medical Center and Children’s Hospital of Richmond with the following exceptions: teams did not visit areas that required handlers to don extra personal protective equipment, rooms where patients tested positive for SARS-CoV-2 (COVID-19), areas where food is served, and active labor and delivery rooms. Table 2 provides a description of each floor and the services provided in those locations.

#### 2.2.8. Location

Interaction location was also recorded. An interaction could take place in a hall (an area not bound by four walls and/or a door including common areas such as lobbies and elevator waiting areas), a patient room (a room designated for patient treatment only), or an office (a room with four walls with a door that designates space for employee functions, elevator, or the volunteer services office.

#### 2.2.9. Hand Sanitizer Use

The Dogs on Call program adheres to infection prevention guidelines recommended by the American Veterinary Medical Association [32] and the Society for Healthcare Epidemiology America [33]. These guidelines indicate that all people who touch a therapy dog should use hand sanitizer before and after each interaction. Handlers are responsible for ensuring that these hand sanitizer guidelines are followed by providing the sanitizer to individuals who wish to interact with the dogs from small bottles they carry with them. Hand sanitizer behavior was coded as before-only (hand sanitizer was used before the interaction), after-only (hand sanitizer was used after the interaction), before and after (hand sanitizer was used before and after the interaction), not applicable (when the participant only talked to a dog or when a patient was unable to pet a dog due to immobility or contact restrictions), or none (when no hand sanitizer was used and physical contact between a human and dog occurred). 

#### 2.2.10. “Drive-Bys”

“Drive-by” interaction definitions were developed for this project by the study team. A drive-by interaction was coded if the interaction lasted 3 s or less and the individual did not pause near the handler or dog. Drive-bys were defined as verbal (talking only), physical (petting only), or both (Talk and Pet). The total number of drive-bys, as well as totals of each type of drive-by (Talk, Pet, Talk and Pet) were recorded for each visit. This definition was developed to distinguish ultra-brief, spontaneous interactions that usually occurred in hallways and other public spaces and were conducted in passing.

### 2.3. Procedure

#### 2.3.1. Data Collection

All visit recipients were either employed by, receiving treatment from, or visiting/accompanying someone at VCU Medical Center. Program protocol requires that visit recipients provide assent before being approached by a therapy dog team. All recipients are free to decline or postpone a visit. As part of the reactivation, the Dogs on Call program implemented extra infection prevention precautions to reduce the transmission of COVID-19 in addition to the program’s standard use of hand sanitizer before and after touching the dog. These precautions included the use of Level 3 face masks and face shields by human handlers, mandatory temperature and respiratory symptom screenings upon entering the hospital and refraining from visiting areas of the hospital that would require donning additional personal protective equipment, such as rooms housing COVID-positive patients or the burn unit. All Dogs on Call handlers are also required to be fully vaccinated against COVID-19 in order to participate in hospital visits. Handlers who did not wish to receive the vaccine were offered the opportunity to participate with their dogs in a virtual visitation program that was not a part of this study. Handler–dog teams are given a maximum time limit of two hours in the hospital to minimize canine stress and fatigue.

The observer accompanied dog/handler teams on visits throughout the hospital during a three-and-a-half-month period between June and September 2021. She walked next to the handler and remained near the handler/dog team throughout their hospital visit. Teams were eligible to visit all inpatient units except those requiring the use of additional personal protective equipment, such as the burn unit, rooms with patients who tested positive for SARS-CoV-2 (COVID-19), and food service areas. Table 2 provides a list of hospital units visited during data collection. The observer met teams in the VS office at the beginning of the visit and followed the team throughout the duration of the hospital visit, ending data collection when the door to volunteer services closed at check-out. The observer documented the details of every human–animal interaction that took place from the time the team clocked in until the time they clocked out. This allowed the observer to observe and record details that are typically only observed by handlers during their visitation time. Data were recorded on an observation sheet in real time, as each interaction took place. Tick marks were used to record visit characteristics on a standardized checklist containing the variables describe above. A stopwatch was attached to the top of the observer’s clipboard for ease of time notation. Given that the data were collected for program quality assurance purposes, researchers did not obtain consent from participants and no personal information was collected from them.

#### 2.3.2. Study Design

The study design was observational and descriptive. The observer made no attempt to engage in teams’ interactions with participants. Handlers were aware of the observer’s role and that interaction characteristics were being recorded for quality assurance purposes.

#### 2.3.3. Data Analysis

All analyses were conducted using SPSS (version 26, IBM, Armonk, NY, USA) and Stata (version 15, StataCorp, College Station, TX, USA). Univariate statistics were used to examine the frequencies and distributions of categorical and continuous variables and to ensure that distributional characteristics of continuous variables were appropriate for planned analyses. 

Role x Time Analysis. The participant role variable was used to obtain a running total of how many members of each group participated in an interaction. Role data were used to create a categorical variable to examine group differences in interaction characteristics. For example, a common interaction consisted of a pediatric patient and their parent/guardian (a public adult). This interaction would fall into the “mixed” category for interactions but would be counted as one pediatric patient and one public adult. A one-way analysis of variance (ANOVA) was performed to examine differences in length of interaction time between individuals of different roles (e.g., pediatric vs. adult patients). Bonferroni’s correction was used to account for multiple comparisons. Post hoc Tukey HSD comparisons were conducted to examine specific group differences in interaction time. 

Gender Differences in Interaction Time. A one-sample, independent groups t-test was conducted to explore gender differences in interaction time. 

Differences in Number of Interactions per Floor. Floor was used as a categorical variable in analyses of interaction characteristics. A chi-squared test using the equiprobability model was used to evaluate differences in the number of visits received by the various floors. Pearson’s standardized residuals were used to determine whether differences in number of visits between floors were statistically significant using a cut-off value of +/− 2.0 [34].

## 3. Results

The observer collected data from 57 visits starting on 2 June 2021 and ending on 15 September 2021. Data were gathered from 69.25 h of observation. There were 1016 interactions recorded, involving 1783 participants. Observed gender information was collected for 1182 participants.

### 3.1. Handler and Dog Characteristics

The majority of handlers (15/20) were female, and all identified as White. Their mean age was 65 years, while the mean age of the dogs was 8 years. On average, dogs were approximately 58.56 cm tall (at the shoulders) and weighed 20.85 kg. Table 3 lists the breeds represented on the teams observed for this study.

### 3.2. Participant Role and Observed Gender

When using participant role as the unit of measure, most visit recipients were healthcare workers (71.69%), and the remainder consisted of 9.30% adult patients, 9.08% pediatric patients, and 9.87% public adults (see Figure 1). When examining interactions as the unit of measure (see Figure 2), 57.2% (581) of interactions occurred with healthcare workers, 12.9% (131) with a mixed population (involving participants from multiple roles), 11.6% (118) with adult patients, 9.6% (98) with pediatric patients, 5.0% (51) with public adults (visitors), and 3.6% (37) with volunteer services staff. No public children were observed. Analyses revealed significant differences in interaction length by participant role (F(4880) = 72.90, *p* = <0.001); post hoc Bonferroni analyses using a *p* value of 0.05/4 showed that patients, both adult (M = 2.58 min, SD = 2.24) (95% C.I = 0.35–1.68) and pediatric (M = 5.81, SD = 4.38) (95% C.I. 3.56–4.97), had longer interaction times than healthcare workers (M = 1.56, SD = 1.92) but not visitors (*p* = 1.00) (see Figure 3). Pediatric patients had significantly longer interaction times than any other group (*p* = 0.001 for all comparisons). 

Observed gender data collection started on 18 July 2021, which was the 24th out of 57 visits. In total, 661 interactions contained information on gender. A total of 928 females (78.40%) participated in canine-assisted interventions. Interactions with females accounted for 64.8% (428) of interactions. Males accounted for 19.1% (126) of interactions, and the remaining interactions involved both genders, accounting for 16.2% (107) of all interactions. Interaction time did not differ significantly by gender (t(552) = −0.736, *p* = 0.462).

### 3.3. Total Visit Time

The average visit time in the hospital for each team was 87 min (SD = 24.69). Total visit times were divided into groups based on 30 min intervals. As illustrated in Figure 4, 16.67% of visits (8) were less than 60 min, 31.25% (15) were 60–90 min, 45.83% (22) were 90–120 min, and 6.25% of visits (3) were more than 120 min. Visit length was missing for 9 visits due to teams arriving earlier than expected or visiting in a pediatric location that does not have a volunteer services office for check-in. Overall, 93.8% of total visit times were less than 120 min (45).

### 3.4. Interaction Characteristics

#### 3.4.1. Interaction Type

The majority of interactions (95.9%) involved a participant both talking to and petting a dog, 2.7% (27) involved a participant only talking to a dog, and 1.5% (15) involved a participant only petting a dog.

#### 3.4.2. Interaction Time

Hospital interaction data were available for all 57 visits. During the visits, a total of 58.97% (40.77 h) of visit time was spent interacting with people. On average, each interaction lasted 2.408 min (SD = 2.96). To better examine the distribution of interaction times, interaction times were divided into one-minute interval groups. As illustrated in Figure 5, 40.26% of interactions (409) were one minute or less, 63.88% (649) were two minutes or less, and 75.49% (767) were three minutes or less. The remaining 24.51% (249) of interactions lasted longer than three minutes. Approximately 2.46% of interactions (25) lasted longer than ten minutes.

#### 3.4.3. Number of People per Interaction

On average, 1.76 (SD = 1.266) people were involved in an interaction (range = 1–14). Over half of interactions (59%) involved one person, 23.7% involved two people, and 17.3% (175) involved three or more people.

### 3.5. Love Bombing

Love bombs made up 16.33% (166) of all interactions. On average, 4.09 people (SD = 1.498) were involved in a love bomb (range = 3–14).

### 3.6. Floors

As shown in Table 4*,* most interactions took place on Inpatient and Inpatient Support (*n* = 286, 29.39%) as well as Critical Care Inpatient floors (*n* = 254, 26.10%). Pediatric Inpatients represented the floor with the third highest number of interactions (*n* = 235, 24.15%).

Chi-squared tests based on the equiprobability model (equal distribution of interactions across hospital floors) were conducted, and Pearson’s residuals were used to examine whether certain floors received significantly greater or fewer interactions than expected. The expected frequency of interactions under conditions of equiprobability in this case was 162 per floor. Results indicated that the observed distribution of interactions differed significantly from that expected under the equiprobability assumption [*χ*^2^ (7) = 381.77, *p* = 0.000], using a significance value of +/−2.00 [33]. As shown in Table 4, results indicated that Inpatient/Inpatient Support, Pediatrics, and Critical Care units received significantly more interactions than other units. In contrast, common areas, non-emergency outpatient centers, and the emergency department received significantly fewer interactions than expected.

### 3.7. Location

Just over half (57%) of interactions (579) occurred in a hallway, 32% of interactions (325) occurred in a patient room, 5.7% of interactions (58) occurred in an office, 1.3% of interactions (13) occurred in an elevator, and 3.9% of interactions (40) occurred in the volunteer services office.

### 3.8. Hand Sanitizer Use

In total, 79.9% of interactions (812) were carried out with proper hand-sanitizing behavior. For 2.7% of interactions (27), hand sanitizer was not required because no physical contact was made with a dog. Hand sanitizer was applied according to protocol (before and after touching a dog) in 77.3% of interactions (785). Participants applied hand sanitizer before touching a dog but not afterwards in 5.3% of interactions (54). In 5.4% of interactions (55), participants applied hand sanitizer after touching a dog, but not before. In 9.4% of interactions (95), no hand sanitizer was applied.

### 3.9. Drive-Bys

A total of 65.8% (669) interactions were considered drive-bys. Approximately 75% (503) of drive-bys were Talk only, 16.74% (112) were Pet only, and 8.07% (54) were Talk and Pet drive-bys. On average, 11.74 (SD = 5.453) drive-bys occurred per visit.

## 4. Discussion

Data from this study provide valuable insights regarding the impact of a hospital-based therapy dog program on patients and healthcare workers, the challenges of implementing therapy animal visitation during the COVID-19 pandemic, and the effectiveness of strategies to maintain animal welfare during a period of intense stress and high demand for therapy animal interventions.

### 4.1. Program Impact on Visit Recipients

Findings suggest that a hospital-based therapy dog program is a highly efficient way to reach large numbers of patients and hospital staff. This may have been particularly important given the potential for increased social isolation faced by patients as a result of restrictions on the number of hospital visitors due to COVID-19. Our data collection period spanned three and a half months (2 June–15 September 2021) and represents observations from 57 therapy dog visits from twenty different therapy dog teams. During these visits, Dogs on Call teams reached a total of 1783 people, including patients, visitors, healthcare workers, and other staff members. Over 70% of visit recipients were healthcare workers; it is important to note that our program reactivated amidst ongoing waves of COVID-19 surges, a period of intense stress for medical providers. A variety of interventions have been mobilized to support those who provide medical care; these include crisis intervention hotlines [17], digital support groups [35], wellness programs [36], emotion regulation training [37], and “nature-inspired recharge rooms” [38]. A therapy dog visitation program may offer unique benefits for busy providers, such as the flexibility of program delivery and stress reduction benefits from ultra-brief interactions [25]. Visits from Dogs on Call can be requested by staff at any time, scheduled for specific employee wellness and stress reduction events, or can occur spontaneously with handler–dog teams deployed throughout the hospital. In addition, our results show that most interactions lasted two minutes or less. This counters the concern that the presence of therapy dogs takes excessive amounts of time and cannot be carried out without significantly impeding clinical care. Further research is needed to evaluate ways in which hospital-based therapy dog programs can be expanded or adapted to address unmet needs among healthcare workers.

Patients and their families may also receive significant benefits from interacting with therapy dogs. Various studies have shown that hospital-based therapy animal programs are associated with a range of physical and mental health benefits for patients, including distraction from pain [39], decreased psychological distress [40], and higher activity levels [41]. Our study highlights that inpatient/inpatient support, critical care, and pediatric units receive significantly more therapy dog visits than other hospital departments. These units care for patients with potentially life-threatening illnesses and injuries. Findings could suggest that healthcare workers and/or volunteer handlers recognize a greater level of need among those patients; further study is needed to determine whether healthcare workers request more therapy dog visits for those patients or whether handlers simply tend to prefer visiting those units.

### 4.2. Strategies for Maintaining Human and Animal Welfare

Findings indicate a high level of adherence to human safety and animal welfare standards. The average age of our human volunteers was 65 years; this age group may be particularly vulnerable to negative sequelae associated with COVID-19 infection, with death rates increasing exponentially with age during the initial outbreak in 2020 and decreasing significantly among this population following introduction of the vaccine [42]. All hospital volunteers were required to receive full doses of the FDA-approved COVID-19 vaccine and provide documentation of vaccination prior to returning to active volunteer status in the hospital, including Dogs on Call handlers. Those handlers who did not wish to receive the vaccine or were unable to receive it due to religious beliefs or pre-existing health conditions were offered the opportunity to participate in a virtual visitation program. Handlers were also required to don Level 3 masks and a face shield or goggles upon entering the hospital to further minimize COVID-19 infection. Hand sanitizer was applied according to protocol in 80% of interactions that involved touching a dog. Additionally, our program asks handlers to systematically track all requested and spontaneous visits by location. Forms designed for this purpose are available when handlers check in for their hospital visits and are entered into a database that can be queried in the event that contact tracing is needed. These strategies were highly effective and no COVID-19 infections were reported among handlers, despite them being present in the hospital during several COVID-19 surges. These infection prevention findings are important given that 96% of interactions in this study involved individuals talking to and petting the therapy dogs in close proximity to handlers.

Our study showed consistent fidelity to animal welfare safety guidelines. The majority (93.8%) of visits were two hours or less, indicating a high level of adherence to animal welfare visit length recommendations. Average visit length for reactivating teams fell under the two-hour time limit required by Pet Partners [43] and the VCU Center for Human–Animal Interaction [4] at 87 min. This visit length was likely shorter than visit lengths under non-pandemic conditions given that teams were instructed to limit their initial three reactivation visits to approximately one hour as their dogs reacclimated to the hospital following their year-long hiatus. Nevertheless, it is encouraging to see that handlers adhered to this recommendation despite the high levels of patient, staff, and healthcare worker need for relief from pandemic-related stress. Anecdotal observations of reactivation visits suggest that people wanted to spend longer amounts of time with the teams, which could have created social pressure for them to remain past the recommended one-hour time limit during reactivation.

### 4.3. Challenges during Program Reactivation

A number of challenges to maintaining high human safety and animal welfare standards presented themselves during our reactivation study. These included “love bombs”, in which groups of people crowded handlers and their dogs, as well as “drive-bys”, interactions in which people would touch the dogs and leave too quickly for handlers to offer hand sanitizer. Although the average number of people per interaction was approximately two people, love bombs could involve as many as fourteen people at once. We developed a “buddy system” to mitigate these departures from protocol; program staff or additional volunteer handlers accompanied teams to assist with crowd control, animal monitoring, and hand sanitizer application. We recommend this system for all unusual situations, such as the activation of a new visitation program, reactivation of an existing program that has been on hiatus, or special events at which crowds of people may congregate.

### 4.4. Visit Recipient Characteristics

Over 75% of visit recipients were female and over 60% of interactions took place among females. An explanation for this difference may lie in the large percentage of women present in healthcare professions such as nursing [44]. There were no significant differences between male and female interaction times, suggesting that men and women are equally willing to engage in human–animal interaction in a hospital. There is a lack of data within other HAI studies that examine spontaneous interactions with therapy dogs across genders. Unlike previous studies, this study was solely observational, where participation in human–animal interaction occurred spontaneously as the dogs and their handlers made themselves available for interactions throughout the hospital. This allowed analysis of what types of participants actively seek therapy dog interactions in a hospital setting. Future research should investigate gender-specific factors that influence the length of engagement with a therapy dog (ex. gender of dog, gender of handler). We also suggest that future researchers examine other issues of diversity, equity, and inclusion to better understand populations served and under-served by hospital-based therapy animal visitation programs.

A major finding of this study was the large percentage of healthcare workers who participated in canine-assisted interaction. Healthcare workers comprised 71.69% of visit recipients and 57.2% of interactions occurred solely between healthcare workers. This may indicate a strong need for stress relief among medical staff due to effects of the COVID-19 pandemic, given that participants voluntarily sought interactions. Recent studies indicate that hospital staff may experience elevated rates of PTSD, suicidal ideation, extreme burnout, and fear during and after a pandemic [20,22,24,45]. The literature shows that interacting with a therapy dog significantly reduces the stress and burnout of healthcare workers [11,25,26]. In-hospital therapy dog programs have potential to significantly combat the negative psychological consequences of COVID-19 among hospital personnel. However, many of these programs were shut down during the pandemic, making utilization of these resources scarce. Our study suggests that hospital-based therapy dog programs may meet stress-reduction needs among hospital staff safely during pandemic conditions. Barker’s (2005) randomized cross-over trial suggests that brief (5 min) interactions with a therapy dog confers similar benefits to longer (20 min) interactions [25]. Additional research is needed to evaluate the effects of ultra-brief (<5 min) therapy dog interactions on stress responses in healthcare workers and to explore the range of these effects more fully.

This study also investigated differences in interaction times by visit recipient roles. Two main findings emerged from this analysis. First, both adult patients and pediatric patients had significantly longer interaction times than other groups. Patients may spend more time with a therapy dog because patients are typically free of time constraints or other work-related obligations. The second finding was that pediatric patients had significantly longer interactions than any other group, including adult patients. Children may spend more time with a dog due to higher levels of stress, greater need for stimulation, or greater levels of excitement when interacting with a dog. Future research should investigate the relationships between age (child vs. adult), hospital role (e.g., healthcare worker, patient, visitor), and stress-related outcomes to maximize the targeted deployment of animal-assisted hospital visitation programs.

### 4.5. Limitations

Several specific limitations warrant consideration when interpreting our findings. First, gender data were not collected until the second month of the study, and given the quality assurance nature of the study, could not be collected by participant interview. Characterizing participants by observation rather than by self-report introduces potential error in gender findings and certainly excludes individuals whose gender identity may not match their physical appearance. Second, the total visit length was unavailable for nine visits; this occurred when the researcher was unable to meet teams directly upon check-in to the hospital. There is no reason to suspect that these nine visits differed in significant ways from visits in which the researcher accompanied the teams from start to finish. Third, recorded interaction times were close approximations rather than precise values and sometimes depended upon the reaction time of the observer during ultra-brief interactions. Relatedly, only one observer counted the number of people involved in interactions, which could have become challenging when crowds formed during love bombs. With that said, error was minimized by having a one-page checklist, recording interaction length using a running timer attached to a clipboard, and using tick marks for counting; these techniques allowed the researcher to rapidly note times and record data with rapid strokes of a pen. In addition, all data were collected by the same person using the same approach; thus, it is reasonable to assume that any measurement error was consistent across interactions and is unlikely to represent a systematic bias in the recording of timings. It is important to point out that video-taping interactions and having multiple raters score each video recording is not possible in this setting due to patient privacy laws and COVID-19-related restrictions on social distancing. Lastly, without testing every handler and visit recipient, definitive conclusions regarding the transmission of COVID-19 (or the lack thereof) cannot be drawn. When these data were collected, rapid testing for SARS-CoV-2 was not widely available and tests were reserved for patients with respiratory symptoms and healthcare workers, rendering daily testing for handlers infeasible. However, temperature checks and respiratory symptom screenings were conducted for all handlers entering the hospital; furthermore, the program’s contact tracing system was consistent with that used by the health system to track potential exposure to the virus.

## 5. Conclusions

The findings indicate that the Dogs on Call hospital-based therapy dog program reached large numbers of patients, staff, and healthcare workers efficiently and safely during successive waves of the COVID-19 pandemic. Strict adherence to human and animal welfare standards allowed the program to serve others with no reported cases of COVID-19 transmission associated with human–animal or human–volunteer contact, despite the highly contagious nature of the virus. To the best of our knowledge, this is the only study of its kind to evaluate such outcomes during the pandemic; the findings can inform policies and procedures for the development and reactivation of other human–animal visitation programs in similar contexts.

## Figures and Tables

**Figure 1 animals-12-01842-f001:**
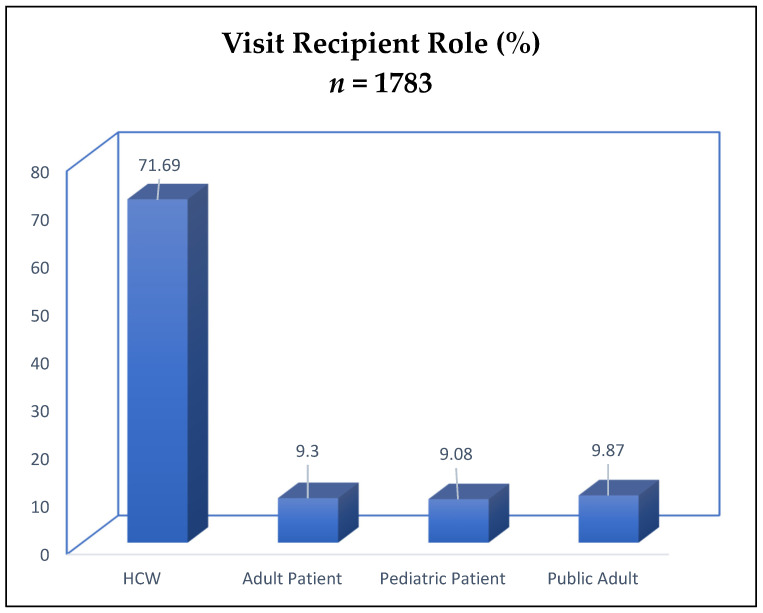
Visit recipient roles as percentage of visit recipients.

**Figure 2 animals-12-01842-f002:**
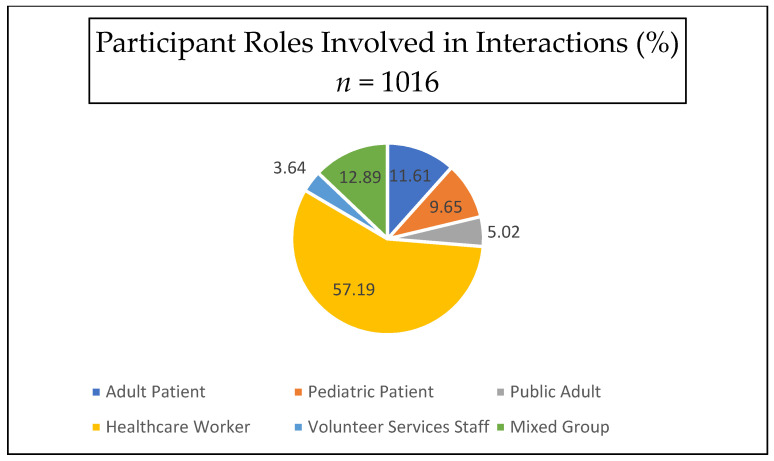
Visit recipient roles as percentages of interactions.

**Figure 3 animals-12-01842-f003:**
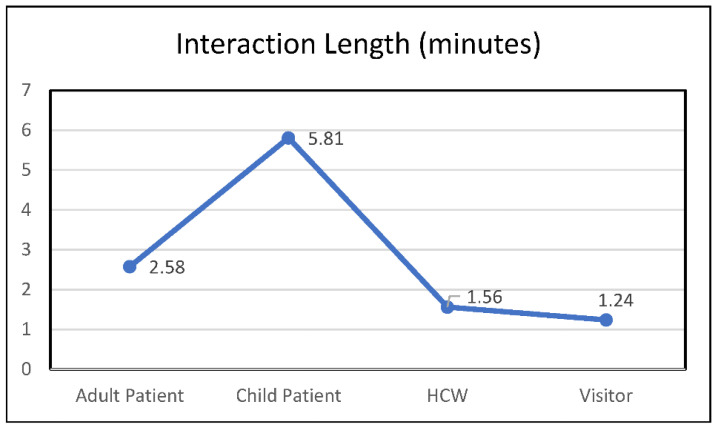
Interaction length by participant role.

**Figure 4 animals-12-01842-f004:**
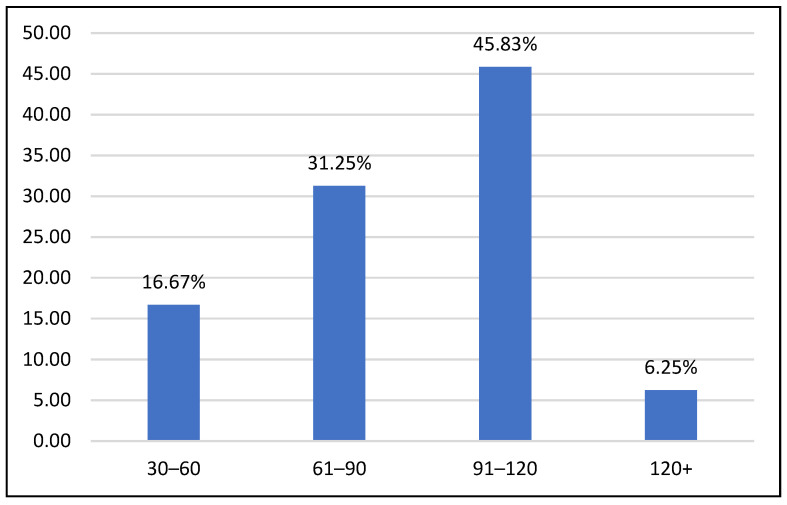
Distribution of total visit times across 30 min intervals.

**Figure 5 animals-12-01842-f005:**
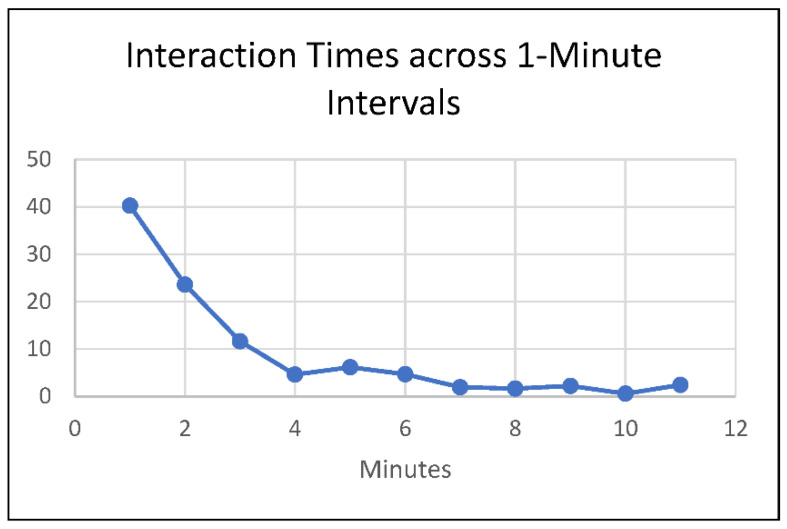
Distribution of interaction times across 1 min intervals.

**Table 1 animals-12-01842-t001:** Handler and dog health and safety requirements.

	Pre-COVID-19	Additions during COVID-19
Handler	Varicella (vaccine or titer)	COVID-19 vaccine
	MMR (measles, mumps, rubella) (vaccine or titer)	COVID-19 booster
	Annual flu vaccine	Level 3 face mask
	Tuberculosis screening	Face shield or goggles
		Temperature measurement and respiratory symptom checklist upon hospital entry
Dog	Registration with Pet Partners or Alliance of Therapy Dogs w/Canine Good Citizen Test	Reactivation shadowing
	Annual veterinary exam	Canine stress evaluation by program staff
	Vaccine or titer for: rabies, distemper, and parvovirus	Three one-hour reactivation visits for reacclimation
	Negative annual fecal exam	
	Two-hour visit limit	
Visit Protocol	Hand sanitizer before/after touching dog	No entry into COVID+ (“Hot”) zones
	Contact tracing	Remain at home if exposed to COVID-19 virus or experiencing respiratory symptoms

**Table 2 animals-12-01842-t002:** Description of hospital floors and services provided.

Common Areas	Areas which all persons in the hospital (staff, visitors, volunteers, etc.) are free to use (with the exception of food service areas where teams do not visit)
Inpatient/Inpatient Support	Floors that provide general medical care and an array of services such as respiratory therapy, trauma treatment, cardiac care, orthopedics, intensive care, etc.
Pediatric	Floors that specialize in the treatment of pediatric patients including the Children’s Hospital of Richmond
ICU	Floors that specialize in the treatment of patients with critical illness or injury
Volunteer Services	Volunteer service office where Dogs on Call teams sign in and out before and after hospital visits
Gateway	The Gateway Building serves as VCU Medical Center’s “front door” and houses some of its outpatient services. Check-in and waiting areas for surgical services are located on the 5th floor of this building
Emergency Department	Department that provides immediate treatment for life threatening or time-sensitive health concerns
Nelson Clinic	Various outpatient services such as OB/GYN & Women’s Health, Outpatient Eye Clinic, and dental care are housed here
West Hospital	West Hospital houses clinical, administrative, and support services for VCU Medical Center, as well as academic and administrative offices of VCU’s School of Medicine and College of Health Professions
Psychiatric, Palliative Care	These departments share the same floor. Psychiatry treats those suffering from mental illness. Palliative care refers to end-of-life treatment

**Table 3 animals-12-01842-t003:** Characteristics of Dogs on Call therapy dogs observed during the quality assurance study *.

Dog	Age (Years)	Sex	Breed	Height (cm)	Weight (kg)
1	-	Female	Labradoodle	-	-
2	4	Female	Golden Retriever	71.12	27.22
3	10	Male	Mixed Breed (Large Terrier/Wolfhound)	78.74	27.22
4	11	Male	English Cream Golden Retriever	71.12	26.76
5	3	Male	English Cream Golden Retriever	71.12	29.94
6	9	Female	Leonberger	88.90	41.73
7	4	Male	Mixed Breed (Terrier x)	30.48	7.26
8	13	Male	Irish Setter	66.04	29.48
9	9	Female	Irish Setter	68.58	29.48
10	10	Female	Pembroke Welsh Corgi	38.10	11.34
11	12	Male	Mixed Breed (Lab/Pug/Boxer)	60.96	21.77
12	7	Female	Golden Doodle	76.20	27.22
13	8	Male	Shih Tzu	38.10	8.16
14	7	Male	Miniature Schnauzer	35.56	3.40
15	13	Female	Jack Russell Terrier	30.48	7.26
16	5	Female	English Cream Golden Retriever	71.12	29.48
17	-	Female	Chocolate Labrador Retriever	-	-
18	8	Male	Standard Wire Hair Dachshund	40.64	12.70
19	10	Female	Maltipoo	25.40	2.27
20	2	Male	English Cream Golden Retriever	91.44	32.66

* Some therapy dog information is missing because although handlers are asked to provide this information, they are not required to do so.

**Table 4 animals-12-01842-t004:** Frequency of interactions by hospital location/floor.

Floor	Interaction Frequency (*n*, %)	Pearson’s Residuals
Common Areas	123, 12.64	−3.08 *
Inpatient/Inpatient Support	286, 29.39	9.72 *
Pediatric Inpatient	235, 24.15	5.72 *
Critical Care	254, 26.10	7.21 *
Non-emergency Outpatient	27, 2.77	−10.61 *
Emergency Department	48, 4.93	−8.97 *
Total	973, 100.0	

* Forty-three interactions were excluded because they occurred in volunteer services or administrative support areas, which do not serve patients.

## Data Availability

Data sharing is not applicable to this article.

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
