# Peer review of "Reactivation of a Hospital-Based Therapy Dog Visitation Program during the COVID-19 Pandemic"

_animals, 2022, doi:10.3390/ani12141842_

Round 1

Reviewer 1 Report

Dear Editor,

Dear authors,

This manuscript uses an observational study design to investigate behaviours towards a dog/human diad when spontaneously entering a hospital. Contact times, contact groups, visited areas etc are investigated. One main finding is the high contact frequency with hospital staff- and this is highly relevant when it comes ways of supporting mental health in helath care workers. In addition, the aim was to investigate a newly adapted visitation protocol to ensure the wellbeing of all two legged and 4 legged parties involved and to find challenges that had to be overcome. This is why I in general support this manuscript to be published. However, some paragraphs mix and match different topics. Some aspects (such as the challenges) only appear in the discussion section; the protocol/the changes to the original protocol per se are not included in detail.

In general it has to be pointed out that no conclusion on the effect of this spontaneous intervention can be drawn when it comes to mental health effects (an observational study without any inerviews or physiological correlates). So it should be emphasised that this is a main future aspect. Are on average 2 minute long interactions truly helping the staff? (given the high number of contacts- and I again assume here that it was not always the same people talking/petting the dog during the visits, it appears so! Interrupting work for a planned intervention is something different for a hospital manager than spontaneous drive by's lasting seconds.

So I recommend focusing more on the actual protocol (pros and cons), findings and challenges (also the issue with the hand sanitiser) rather than speculating too much about mental health implications.

Please see some details:

Abstract:

Minimising contact to people and other safety measures were one reason why HAI were stopped. Please rephrase your abstract to elaborate on the safety measures and challenges in more details e.g. what was difficult to apply ? what were those 20% and the people aggregations?

Introduction:

Please elaborate on the physiological correlates (measurement tools) so that readers can understand how these improvements were measured for example serum cortisol reduction in the study by Barker et al. 2005; questionnaires by Jensen et al. 2021)

There is a recent systematic review that should be included in the introduction: Acquadro Maran, D., Capitanelli, I., Cortese, C. G., Ilesanmi, O. S., Gianino, M. M., & Chirico, F. (2022). Animal-Assisted Intervention and Health Care Workers’ Psychological Health: A Systematic Review of the Literature. Animals, 12(3), 383.

Line 80ff 1.1. Barriers to Program Implementation

This paragraph comprises both; program implementation issues (having protocols per se, disease transmission in general but also the rather unregulated field of assistance vs therapy dogs- training, mandatory health checks of the dogs etc) and program continuation issues (due to the pandemic)

Please disentangle these issues within your paragraph.

M&M

Line 137 2.1. Therapy Dog/Handler Teams

You clearly articulate how important safety measures and health screenings are. Also you indicate that a “new” protocol was used (line 113). Please provide this information in more detail;  you are mentioning specific health screenings and vaccinations for the handler (what are they?) and that the dog needs to be registered by an organization; but what health screenings do the dogs need to have? What were your new measures in your safety protocol (I assume covid testing of the handlers was one of them or only the covid vaccination??; any social distancing measures human to human?)

One of your aims is to help with these aspects (line 129) so please provide them.

Some of your covid measures are written under 2.3.1. data collection- this is unclear to me- why there? that clearly needs to be in your safety protocol (you don’t have a subheading for that yet); also some aspects have been explained earlier on already e.g. drive by’s; interaction location , stopwatch on the clipboard….

Line 195 could you please clarify for me where this observer was located? Right next to the dog/handler team? Watching from the distance?

Line 241 before needs to be replaced with after

Results

All Tables and figures: the quality appears to me as screenshot quality (blurred lines)- please improve.

Line 426: hand sanitizer use, could you please elaborate if there where specific groups that used or did not use hand sanitizer; for example I struggle to understand from where a visitor should know your regulation when it comes to hand sanitizer use (before and after touching the dog) and from where they would get hand sanitizer from? Also “drive by’s and "love bombs” seem to me the obvious non adhering to regulations situations? Is that true? Did you look into that- where is the data?

discussion

Line 510: “to remain past the recommended one-hour time limit during reactivation” this is written confusingly; the recommendation for reactivation is 1 hour, the average visitation was 2 hours; so you cannot use this to argue about maintaining dog welfare…..

Line 514ff challenges: “these included "love bombs",in which groups of people crowded handlers and their dogs as well as "drive bys", inter-
actions in which people would touch the dogs and leave too quickly for handlers to offer hand sanitizer.

As already mentioned in results: do you have data to collaborate this? I expected them to be issues, but it is only in the discussion that I read about this

Because this is then one of your main findings when I read into your aims for this study….. the challenges you encountered with your new protocol and the techniques you used such as the “buddy system” ; these challenges should then be included in the abstract

What do hospitals need to proactively look out for when reimplementing HAI?

Line 551: your study has a  much lower contact time than previous studies; you do not know if brief contacts such as drive by’s truly reduce stress (you did not ask people, or measured anything). I completely agree, the amount of contact with health care professionals is highly relevant, especially given that these were not scheduled visits (like in another studies). So to me the question is: Are unplanned and spontaneous, short time (less than 2 minutes and even seconds) contacts, where health care workers do not need to interrupt their work due to a scheduled dog appointment (see details Kline et al. 2020) also efficient in improving mental health? You should discuss this point in more detail.

Line 557: longer time of patients spent with dog might also be due to the access limitation of general visitors to the hospital? So patients were socially more isolated and hence a greater need for contact with a dog? (at least here, visitor numbers were highly controlled and only a limited number of people and also time spent in the hospital were allowed)

Limitations- I imagine it to be really difficult to “measure” a love bomb with up to 14 people simultaneously with one stop watch; also please specify that you cannot infer any changes in a stress response, mental wellbeing improvement etc. given the observational study design;

Also if your handlers were not tested for corona virus before entering the hospital, you do not know if the handlers transmitted the disease to someone else in the hospital; this should be a safety measurement in every standard protocol (if you did not do it, then at least a recommendation to improve your protocol)

Author Response

Reviewer #1:

Abstract:

Minimising contact to people and other safety measures were one reason why HAI were stopped. Please rephrase your abstract to elaborate on the safety measures and challenges in more details e.g. what was difficult to apply ? what were those 20% and the people aggregations?

Response: We found it difficult to provide a thorough elaboration of these details in a 200-word abstract. We added brief descriptions of the safety measures, challenges to their implementation, and mitigation measures to the abstract - these can be found on lines 26-28 and 37-40.

Introduction:

Please elaborate on the physiological correlates (measurement tools) so that readers can understand how these improvements were measured for example serum cortisol reduction in the study by Barker et al. 2005; questionnaires by Jensen et al. 2021)

Response: We did not collect physiological data. We have added the observation checklist as an Appendix to the article.

There is a recent systematic review that should be included in the introduction: Acquadro Maran, D., Capitanelli, I., Cortese, C. G., Ilesanmi, O. S., Gianino, M. M., & Chirico, F. (2022). Animal-Assisted Intervention and Health Care Workers’ Psychological Health: A Systematic Review of the Literature. Animals12(3), 383.

Response: We have added a citation of this paper in the Introduction on lines 78-80 and 759-761 (References).

Line 80ff 1.1. Barriers to Program Implementation

This paragraph comprises both; program implementation issues (having protocols per se, disease transmission in general but also the rather unregulated field of assistance vs therapy dogs- training, mandatory health checks of the dogs etc) and program continuation issues (due to the pandemic)

Please disentangle these issues within your paragraph.

Response: We re-wrote these paragraphs so the concepts therein are clearer. These changes occur on lines 84-124.

M&M

Line 137 2.1. Therapy Dog/Handler Teams

You clearly articulate how important safety measures and health screenings are. Also you indicate that a “new” protocol was used (line 113). Please provide this information in more detail;  you are mentioning specific health screenings and vaccinations for the handler (what are they?) and that the dog needs to be registered by an organization; but what health screenings do the dogs need to have? What were your new measures in your safety protocol (I assume covid testing of the handlers was one of them or only the covid vaccination??; any social distancing measures human to human?)

One of your aims is to help with these aspects (line 129) so please provide them.

Response: We created a table that details pre- and during COVID health and safety requirements for handlers and dogs. This table can be found on lines 168-171.

Some of your covid measures are written under 2.3.1. data collection- this is unclear to me- why there? that clearly needs to be in your safety protocol (you don’t have a subheading for that yet); also some aspects have been explained earlier on already e.g. drive by’s; interaction location , stopwatch on the clipboard….

Response: We have changed the location of these measures and placed them in a table located on lines 168-171.

Line 195 could you please clarify for me where this observer was located? Right next to the dog/handler team? Watching from the distance?

Response: We added a description of the observer's location on lines 307-308 in the Data Collection section.

Line 241 before needs to be replaced with after

Response: We made this correction on what is now line 275.

Results

All Tables and figures: the quality appears to me as screenshot quality (blurred lines)- please improve.

Response: We have provided a separate document that contains the images. We have been unable to improve the quality of the images while fitting them into the template. We hope that the publishing staff can help us with this.

Line 426: hand sanitizer use, could you please elaborate if there where specific groups that used or did not use hand sanitizer; for example I struggle to understand from where a visitor should know your regulation when it comes to hand sanitizer use (before and after touching the dog) and from where they would get hand sanitizer from? Also “drive by’s and "love bombs” seem to me the obvious non adhering to regulations situations? Is that true? Did you look into that- where is the data?

Response: We did not collect finely grained data from visit recipients that would allow us to test associations between visit recipient role and use of hand sanitizer given that this was an IRB-exempt, quality assurance study.

discussion

Line 510: “to remain past the recommended one-hour time limit during reactivation” this is written confusingly; the recommendation for reactivation is 1 hour, the average visitation was 2 hours; so you cannot use this to argue about maintaining dog welfare…..

Response: We added language to clarify that the mean length of hospital visits was likely shorter than usual given that handlers were instructed to limit their initial three reactivation visits to one hour (lines 567-568).

Line 514ff challenges: “these included "love bombs",in which groups of people crowded handlers and their dogs as well as "drive bys", inter-
actions in which people would touch the dogs and leave too quickly for handlers to offer hand sanitizer.

As already mentioned in results: do you have data to collaborate this? I expected them to be issues, but it is only in the discussion that I read about this

Because this is then one of your main findings when I read into your aims for this study….. the challenges you encountered with your new protocol and the techniques you used such as the “buddy system” ; these challenges should then be included in the abstract

What do hospitals need to proactively look out for when reimplementing HAI?

Response: We added a description of challenges to the abstract. In terms of data to support the observations of "love bombs" and "drive-bys", these data are presented in the Results section on lines 447-448 and 494-496. Specifically, 16.33% of interactions met the definitional criteria for love bombs with an average of 4.09 people involved; 65.8% of interactions met definitional criteria for "drive-bys", with an average of 11.74 occurring per visit. We also recommended a "buddy system" that programs can use for crowd control and to improve the use of hand sanitizer among crowds. That recommendation appears on lines 581-586.

Line 551: your study has a  much lower contact time than previous studies; you do not know if brief contacts such as drive by’s truly reduce stress (you did not ask people, or measured anything). I completely agree, the amount of contact with health care professionals is highly relevant, especially given that these were not scheduled visits (like in another studies). So to me the question is: Are unplanned and spontaneous, short time (less than 2 minutes and even seconds) contacts, where health care workers do not need to interrupt their work due to a scheduled dog appointment (see details Kline et al. 2020) also efficient in improving mental health? You should discuss this point in more detail.

Response: We added additional discussion of the need for future research on this point on lines 615-620.

Line 557: longer time of patients spent with dog might also be due to the access limitation of general visitors to the hospital? So patients were socially more isolated and hence a greater need for contact with a dog? (at least here, visitor numbers were highly controlled and only a limited number of people and also time spent in the hospital were allowed)

Response: We did not measure the degree of social isolation for patients as this was a quality assurance study. We added a point about the unusual nature of pandemic conditions in the hospital with specific reference to the likelihood of increased social isolation due to visitor restrictions on lines 507-509.

Limitations- I imagine it to be really difficult to “measure” a love bomb with up to 14 people simultaneously with one stop watch; also please specify that you cannot infer any changes in a stress response, mental wellbeing improvement etc. given the observational study design;

Also if your handlers were not tested for corona virus before entering the hospital, you do not know if the handlers transmitted the disease to someone else in the hospital; this should be a safety measurement in every standard protocol (if you did not do it, then at least a recommendation to improve your protocol)

Response: We added a sentence describing the limitations of counting the number of people involved in an interaction using one observer on lines 644-645.

When these data were collected, rapid testing for the SARS-CoV-2 virus was reserved for patients with respiratory symptoms and healthcare workers, rendering daily testing for handlers infeasible. We added this to the limitations section and discussed the temperature and respiratory symptom checks conducted for handlers whenever they entered the hospital. This can be found on lines 654-661.

Reviewer 2 Report

No concerns noted. Authors clearly explained and justified the value and applicability of their findings, beyond the COVID-19 pandemic. Explanations were provided for any potential deviations from protocol (ex, gathering info re: observed gender).

Author Response

Reviewer 2: No changes requested. Thank you for your review.

Reviewer 3 Report

 This is a very interesting manuscript exploring a very novel area of research. Very well-structured introduction with a strong rationale for the purpose of the study. The authors highlight the importance of this research in order to facilitate making well-informed decisions regarding implementation of in-hospital therapy dog programmes. This is very valuable and a timely contribution to the field, given there is such a dearth of literature around the provision of AAIs during Covid-19.

The authors acknowledge the difficulty in drawing generalisable conclusions due to the variability in execution of HAI programmes in hospitals. This is a well-established challenge and as such, it is valuable to see the current manuscript including specific detail about the intervention delivered (e.g., specific characteristics about dogs, location, etc).

Minor comments:

1.     The authors note the protocol was deemed exempt from ethical review. At this point, I wondered why as a reader as the programme was being delivered to potentially vulnerable participants and the researcher was observing the delivery. However, I read from line 285 that this was for quality assurance purposes. Please could this also be included on lines 133-135 for clarification earlier on.

2.     Where interaction characteristics were recorded on a pre-defined checklist, was this pre-defined by the research team or based on existing literature? Please could this just be clarified.

3.     Line 213, ‘love bombing was defined as…’ was this defined by the team or has this been defined elsewhere? Upon reading further, I noted the authors specified that ‘drive-bys’ were interactions developed for the project by the team. It would be beneficial to include this detail for ‘love bombing’.

I look forward to seeing this paper published.

Author Response

Reviewer 3:

  1. The authors note the protocol was deemed exempt from ethical review. At this point, I wondered why as a reader as the programme was being delivered to potentially vulnerable participants and the researcher was observing the delivery. However, I read from line 285 that this was for quality assurance purposes. Please could this also be included on lines 133-135 for clarification earlier on.

Response: We added this clarification on lines 152-154 and expanded upon the reasons why the study met IRB criteria for exemption.

  1. Where interaction characteristics were recorded on a pre-defined checklist, was this pre-defined by the research team or based on existing literature? Please could this just be clarified.

Response: We added a sentence indicating the origins of the checklist (human-animal interaction expertise of the third author [N.R.G.] and feedback from volunteers regarding their experiences in the hospital). This can be found on lines 215-217.

  1.  Line 213, ‘love bombing was defined as…’ was this defined by the team or has this been defined elsewhere? Upon reading further, I noted the authors specified that ‘drive-bys’ were interactions developed for the project by the team. It would be beneficial to include this detail for ‘love bombing’.

Response: We added a sentence clarifying that the term "love bomb" was coined by the authors based on initial observations of reactivation visits. This can be found on lines 243-244.